# Quantification of Signal Transduction Pathway Activity in Pancreatic Ductal Adenocarcinoma

**DOI:** 10.3390/ijms262311385

**Published:** 2025-11-25

**Authors:** Léon Raymakers, Lois A. Daamen, Martijn P. W. Intven, Jeanette H. W. Leusen, Patricia A. Olofsen, Anja van de Stolpe, Reinier A. P. Raymakers

**Affiliations:** 1Center for Translational Immunology, University Medical Center Utrecht, Heidelberglaan 100, 3584 CX Utrecht, The Netherlands; l.raijmakers-3@umcutrecht.nl (L.R.); j.h.w.leusen@umcutrecht.nl (J.H.W.L.); 2Division of Imaging & Oncology, University Medical Center Utrecht Cancer Center, Heidelberglaan 100, 3584 CX Utrecht, The Netherlands; 3Department of Surgery, Regional Academic Cancer Center Utrecht, University Medical Center Utrecht Cancer Center & St. Antonius Hospital Nieuwegein, Heidelberglaan 100, 3584 CX Utrecht, The Netherlands; 4Department of Radiotherapy, Division Imaging and Oncology, University Medical Center Utrecht, Heidelberglaan 100, 3584 CX Utrecht, The Netherlands; m.intven@umcutrecht.nl; 5Drug Companion Diagnostics Company (DCDC-Tx), Eikenheuveldreef 16, 5263 EM Vught, The Netherlands; anja.van.de.stolpe@dcdc-tx.com

**Keywords:** signal transduction pathways, STPs, MAPK, pancreatic ductal adenocarcinoma, PDAC, simultaneous transcriptome-based activation profiling, STAP-STP technology, differentiation therapy

## Abstract

Patients with pancreatic ductal adenocarcinoma (PDAC) face a very poor prognosis despite advancements in therapeutic strategies. Signal transduction pathways (STPs) that show altered activity in cancer cells may provide new therapeutic targets. Here, we used simultaneous transcriptome-based activation profiling (STAP)-STP technology to identify abnormal STP activity in PDAC. STAP-STP infers STP activity from messenger RNA expression of the target genes of each pathway-associated transcription factor, which is not possible with conventional bioinformatic analysis. We searched the Gene Expression Omnibus database for publicly available PDAC Affymetrix (GPL570) datasets and included six datasets: four datasets with samples from both normal pancreatic duct epithelial cells and PDAC tumor cells and two datasets with PDAC derived cell lines. The activity of the twelve most relevant STPs (androgen receptor, estrogen receptor, PI3K, MAPK, TGFβ, Notch, Hedgehog, Wnt, NFκB, STAT1/2 type I interferon, STAT1/2type II interferon and STAT3) was quantified. Increased activity of the MAPK, STAT3, Wnt, Hedgehog, Notch TGFβ, and NFκB pathways was found in at least two out of four datasets. In PDAC cell lines, MAPK, PI3K, and STAT3 STPs showed higher activity than in patient samples. Cell type deconvolution analysis showed a variable mixture of fibroblasts, immune cells, and tumor cells in the patient samples, which likely influenced the STP activity profile. This is the first time that STP activity has been quantified in PDAC. We conclude that PDAC is characterized by increased MAPK STP activity in combination with high Ki67 and increased activity of developmental pathways (Wnt, Hedgehog, Notch, TGFβ). Drugs targeting specific STPs will be evaluated in PDAC model systems to develop new therapies for PDAC.

## 1. Introduction

Pancreatic ductal adenocarcinoma (PDAC) has one of the lowest 5-year survival rates of all cancers and is projected to become the second-leading cause of cancer-related death by 2030 [1,2,3]. Poor outcome for patients with PDAC can be attributed to the late onset of symptoms, rapid progression, metastatic potential, and high resistance to anticancer therapies [4,5]. Despite advances in conventional anti-cancer therapies, breakthroughs are not forthcoming [5]. Therefore, the development of new treatment strategies is essential to improve outcomes for patients with PDAC.

The introduction of targeted therapy has improved outcomes in many cancers [6], but for PDAC, this has not yet been achieved [7,8]. The two most used targeted therapies are monoclonal antibodies and small-molecule inhibitors that act on growth factors, cell surface receptors, or intracellular proteins [6]. These drug targets are part of signal transduction pathways (STPs). The activity of STPs is regulated by ligand-receptor protein interactions and determines cellular functions such as division, differentiation, and survival. Genomic, epigenetic, and environmental changes in cancer cause the aberrant activity of STPs, which drives tumor cell proliferation and metastasis [9,10,11]. These pathways can be drug targets for specific therapeutic interventions if their activity can be quantified [9]. However, measuring STP activity has always been challenging due to STP complexity and crosstalk between pathways [9].

Simultaneous transcriptome-based activation profiling of STP (STAP-STP) is an innovative technology to quantify STP activity. The quantification of STP activity is based on expression levels of a defined set of direct messenger RNA (mRNA) target genes of the transcription factors involved in the specific pathway and has been validated in previous studies [12,13,14,15,16,17]. The quantification of STP activity is not possible with conventional bioinformatic analysis [9]. In this study, publicly available Affymetrix microarray transcriptome data were retrieved from the Gene Expression Omnibus (GEO) database. We compared the activity of the twelve most relevant STPs of normal pancreas samples with primary PDAC tumor samples. Alterations in STP activity play a role in tumor growth and metastasis and pose opportunities for therapeutic interventions [18]. Second, we compared the STP activity of primary PDAC tumor samples with PDAC tumor cell lines to identify which cell lines are the best to use for therapy development.

## 2. Results

The STP activity profile of PDAC was determined in four studies from the GEO database containing mRNA expression data. Two datasets included a total of 52 paired samples, enabling comparison between tumor cells and adjacent normal epithelial cells [19,20]. Two other datasets contained a total of 38 tumor cell samples and 15 normal pancreatic tissue samples from non-pancreatic disease donors (Figure 1) [21,22].

### 2.1. STP Activity Profile of PDAC

Since more than 90% of PDAC patients have acquired an activating Kirsten rat sarcoma virus oncogene homolog (KRAS) gene mutation, resulting in activation of the mitogen-activated protein kinase (MAPK) pathway, we expected an increase in the activity of this STP [8,23,24,25]. As expected, increased MAPK STP activity was found in PDAC in all datasets (Figure 1A). MAPK activity was positively correlated with expression of the proliferation marker gene Ki67 (Figure 1B).

Downstream activation of the phosphoinositide 3-kinases (PI3K) pathway can be a consequence of an activating KRAS mutation [25]. PI3K STP activity is inversely related to Forkhead box O (FOXO) transcription factor activity. Independent of PI3K, oxidative stress induces FOXO activity (13). For that reason, the expression of the oxidative stress target gene superoxide dismutase 2 (SOD2) was also determined (Figure 2A). FOXO activity was increased in combination with increased SOD2 gene expression (Figure 2B and Appendix A) only in the PDAC samples of GSE15471. When analyzing individual samples, there was a strong positive correlation between FOXO activity and SOD2 expression in PDAC samples from all datasets (Appendix A). FOXO activity was not inversely correlated with Ki67 gene expression, a measure of proliferation activity (Figure 2C). Together, these results suggest that high FOXO activity was caused by oxidative stress. Consequently, PI3K STP activity could not be interpreted.

Besides MAPK, the activity of the signal transducer and activator of transcription 3 (STAT3), Hedgehog (HH), Notch, transforming growth factor beta (TGFβ), Wingless related integration site (Wnt), and nuclear factor kappa B (NFκB) signaling pathways was significantly increased in two or more datasets (Figure 3). Androgen receptor (AR) and STAT1/2 Type II interferon STP activity was increased in only one dataset (Appendix A). Estrogen receptor (ER) STP activity was not altered (Appendix A). JAK-STAT1/2 type I interferon STP activity was decreased in only one dataset (Appendix A).

Wnt and Notch STP activity have been described to be either simultaneously upregulated or inversely correlated in many cancers [26,27]. We did not find a correlation between Wnt and Notch STP activity in PDAC (Figure 4A and Appendix A). In pancreatic cancers, MAPK and TGFβ STPs are simultaneously activated and contribute to oncogenesis and disease progression [28]. A positive correlation between MAPK and TGFβ STP activity was indeed found in the analyzed datasets (Figure 4B and Appendix A). As illustrated in Table 1, STAP-STP analysis uniquely enables the calculation of the correlation between MAPK and TGFβ STP activity for individual PDAC samples and their normal controls. Furthermore, the MAPK STP has been described to interact with Wnt, Notch, and HH STPs in various cancers [29,30,31]. In the PDAC samples, MAPK STP activity was positively correlated with Wnt and Notch STP activity (Figure 4C and Appendix A).

### 2.2. Purity of the Cell Samples

We analyzed the presence of non-epithelial cells in the samples. In the cell line samples, nearly 100% of the cells were identified as epithelial lineage, confirming the validity of the deconvolution software (Figure 5). Fibroblasts and immune cells were present in variable percentages in all samples (Figure 5). The average epithelial cell and fibroblast content in the tumor cell samples ranged from 33–45% and 45–60%, respectively (Figure 5). Around 5–10% of the cells in tissue samples (both control and PDAC) were immune cells (Figure 5). Interestingly, laser microdissection in GSE17891 showed an average of only 50% epithelial cells (Appendix A). This dataset was not included in the STP analysis since it lacks normal controls. The immune cells were further analyzed for the presence of specific immune cell subsets (Appendix A). Granulocytes showed the highest prevalence in most samples (Appendix A).

Cell sample purity was further analyzed by determining the gene expression of epithelial markers or tumor-associated antigens (TAAs) frequently overexpressed in PDAC [32]. The TAAs EpCAM, TROP2, and MUC1 showed higher gene expression levels in PDAC and were positively correlated with the epithelial cell content as measured by the deconvolution software (Appendix A). MAPK STP activity positively correlated with TROP2 gene expression, while a positive trend was observed for MAPK STP and MUC1 gene expression. This suggests that increased activity of the MAPK STP is predominantly present in the epithelial compartment of PDAC, in line with the presence of MAPK-activating mutations. NFκB STP activity negatively correlated with EpCAM (Appendix A), suggesting higher NFκB STP activity in the non-epithelial cells, most likely in immune cells.

### 2.3. Validity of Tumor-Adjacent Normal Tissue as a Control

Cancer can influence tumor-adjacent normal tissue via mechanisms such as paracrine signaling and by inducing inflammation [33]. For this reason, we compared the STP activity between tumor-adjacent ‘normal’ tissue and normal donor pancreas (Figure 6). Because ER STP activity is similar between normal tissue and PDAC (Appendix A), it could be used as a reference. GSE71989 and GSE16515 showed similar ER STP activity, allowing direct comparison between STP activity scores (Figure 6A). TGFβ STP activity was higher in tumor-adjacent tissue than in normal pancreas (Figure 6B). The other STPs showed no significant alterations in pathway activity (Figure 6C and Appendix A), showing the validity of tumor-adjacent normal tissue as a control.

### 2.4. The STP Activity Profile of PDAC Cell Lines

The STP activity of PDAC-derived cell lines was compared with the patient samples to determine which cell lines best resemble patient PDAC and are preferred as a model for drug development (Figure 7 and Appendix A) [34,35]. On average, cell lines showed higher activity of the MAPK and STAT3 proliferation pathways (*p* < 0.0001) (Figure 7A), which is also reflected in increased Ki67 expression (*p* < 0.0001) (Figure 7B). The low FOXO activity of cell lines indicates high PI3K STP activity (Figure 7A). The Notch, TGFβ, and NFκB pathways showed lower pathway activity in the cell lines relative to the tumor samples (*p* < 0.0001), while other pathways had similar activity (Figure 7A). No clear differences in STP activity between primary and metastatic-derived PDAC cell lines were observed, except for, on average, slightly lower TGFβ activity in the metastasis-derived cell lines (*p* = 0.0045) (Figure 7 and Appendix A).

## 3. Discussion

STAP-STP technology was applied to quantify the activity of the twelve most relevant STPs in PDAC to establish targets for therapy. This study shows that, compared with normal tissue, the STP profile of PDAC is characterized by increased activity of the proliferation pathways MAPK and STAT3, the developmental pathways HH, Notch, TGFβ, and Wnt, and the immune pathway NFκB. MAPK activity in individual tumor samples was positively correlated with activity of the developmental pathways TGFβ, Wnt and Notch. Only correlation with the TGFβ pathway has been described before in PDAC [28]. HH STP activity showed no correlation with MAPK STP activity. This can be explained by high HH activity in non-tumor cells in the samples, such as fibroblasts. Indeed, HH STP activity is known to be high in certain subsets of cancer-associated fibroblasts present in PDAC [36]. The increased activity of developmental pathways is characteristic of the stem cell-like phenotype of PDAC. High activity of the developmental pathway TGFβ may play a role in epithelial-mesenchymal transition (EMT) due to the high prevalence of SMAD gene mutations in PDAC [18,37]. We found that PDAC-derived cell lines showed, on average, higher activity of proliferative pathways and lower activity of the Notch, TGFβ, and NFκB pathways when compared to primary PDAC samples.

The PI3K pathway has been described as abnormally active in PDAC and has been defined as a target for therapy [25,38,39]. Because of cellular oxidative stress in both normal pancreatic tissue and PDAC, the activity of the PI3K STP could not be reliably inferred from FOXO activity (Figure 2A) [13]. In the PDAC cell lines, however, low FOXO activity indicated high PI3K activity.

The datasets GSE15741, GSE16515, and GSE71989 showed similar increased STP activity. Such an increase was not always observed in GSE32676, which can be explained by the difference between the controls. The normal controls of GSE32676 showed the lowest percentage of epithelial cells (on average, 25%) and high activity of HH, TGFβ, and NFκB. It is known that fibroblasts and immune cells have high HH, TGFβ, and NFκB activity [36,40,41]. In contrast to the normal controls, the tumor samples of GSE32676 showed comparable sample purity and STP activity to the other datasets. Our study relies on these previously generated datasets, which likely differ in methodology and quality, and STP activity should be confirmed in future experiments when testing drugs interfering with these pathways.

PDAC tumors are characterized by a relatively low number of tumor cells and a dense desmoplastic stroma, which can make up >90% of the tumor mass [5,42]. All samples, even normal pancreatic tissue, show high percentages of fibroblasts. The control samples of GSE71989 and the tumor samples of GSE16515 showed high epithelial cell purity with low cell type variation and might therefore have been the most accurate datasets. This was only minimally improved in the datasets with laser microdissected samples [43]. A limitation of the deconvolution software is that it is based on RNA expression, which may differ from protein expression.

Overall, STP activity of the normal pancreas and tumor-adjacent tissue controls was quite similar for most pathways. Tumor-adjacent normal tissue samples showed higher TGFβ activity, which could be explained by the presence and the enhanced TGFβ activity of fibroblasts [40]. In line with this, the tumor-adjacent control samples included more fibroblasts. Another explanation could be the effect of the tumor on adjacent normal tissue, since TGFβ activity plays an important role in PDAC [40]. NFκB STP activity showed a trend towards higher activity in tumor-adjacent tissue. This might be due to a tumor-induced inflammatory response that can extend beyond the borders of the tumor [40,41].

The proliferative pathways MAPK, STAT3, and PI3K showed higher activity in the cell lines. This might be explained by the admixture of fibroblasts and immune cells in the tumor samples. Another explanation can be clonal selection during the generation of the cell lines or culture conditions. The lower activity of TGFβ and NFκB STPs in the cell lines can be explained by the absence of fibroblasts and immune cells in the cell line samples [40,41]. Some cell lines were evaluated in two independent datasets, and variation in STP activity was observed. These variations can be explained by differences in culturing conditions and passage number, since Affymetrix expression has been proven to be highly reproducible [44]. STP activity of cell lines should be verified with consistent culturing conditions, and the influence of serum concentration in the culture medium should be evaluated.

STAP-STP analysis provides unique insights into the abnormal functioning of signal transduction pathways in PDAC. MAPK and PI3K are probably STPs that drive tumor growth, given that more than 90% of PDAC tumors have KRAS mutations. TGFβ may be another important pathway, given the high mutation rate in SMAD genes [37]. Interfering with specific STPs and quantifying the influence on all STPs provides evidence that STPs drive tumor growth and how these pathways crosstalk. Correlating STP activity with clinical outcomes, such as disease-free interval and overall survival, increases the probability that these pathways are involved in proliferation and strengthens the rationale to interfere in these pathways.

The abnormally active STPs are in line with previously described pathways that play a role in PDAC [7,8,11]. This is the first time STP activity has been quantified for PDAC, as this cannot be performed with conventional bioinformatic analysis methods, as has been described before [9]. The original publications of the four included GEO datasets describe increased mRNA levels of proteins that play a role in MAPK and PI3K proliferation pathways and are involved in EMT [19,20,21]. One study did not evaluate proteins involved in signaling pathways at all [22]. This confirms the unique approach of the STAP-STP technology applied in this paper.

The identified aberrant STP activity in PDAC can be used for therapy development. In previous studies, targeted therapies have been tested for MAPK, PI3K, STAT3, HH, Notch, TGFβ, Wnt, and NFκB pathways [7]. Most single-drug therapies appeared ineffective. Especially the MAPK and PI3K pathways crosstalk, and inhibition of MAPK, for example via KRAS inhibitors, is often negated by upregulation of Akt in the PI3K pathway [7]. STAP-STP technology is unique in monitoring the influence of drugs on all STPs and identifying crosstalk, resulting in therapy resistance and helping to develop combinational therapy. It also enables the development and evaluation of cancer differentiation therapy by affecting all relevant pathways. An advantage of such an approach is that differentiated tumor cells will become more visible to the immune system [45].

In conclusion, this study showed that PDAC is characterized by a quantified increase in activity of MAPK STP and of the developmental pathways Wnt, HH, Notch, and TGFβ, suggesting a stem cell character caused by EMT. This STP analysis, based on GEO database data, provides information that is essential for the development and evaluation of new therapies, including cancer differentiation therapy.

## 4. Materials and Methods

### 4.1. Affymetrix GEO Datasets

We searched the GEO database (RRID:SCR_005012) for Affymetrix (GeneChip™ Human Genome U133 Plus 2.0 Array, GPL570, RRID:SCR_007817) datasets from studies on PDAC (https://www.ncbi.nlm.nih.gov/gds/). Datasets were included from treatment-naïve patients who underwent surgical resection for PDAC. We limited our analysis to datasets that included ‘normal’ pancreas tissue as a reference. Normal tissue was either tumor-adjacent non-cancerous tissue (paired with a tumor sample) or pancreas tissue from non-pancreatic disease (unpaired) (Table 2) [19,20,21,22,34,35]. The tumor regions were microscopically selected and dissociated with no further tumor cell enrichment, but the exact method was not always specified. GSE17891 included 20 cell lines and 27 PDAC tumor samples enriched for tumor cells via laser microdissection. However, the tumor samples from this dataset were not included in the final analysis, as they lacked normal tissue controls. GSE17891 and GSE21654 include data from 31 unique cell lines, with 17 derived from primary PDAC and 14 from metastatic disease (Table 2). Two datasets (GSE18670 and GSE22780) were excluded from the analysis since all samples failed quality control [46,47].

### 4.2. Measurement of Signal Transduction Pathway Activity

STAP-STP technology was used to quantify pathway activity of the following STPs: the AR, ER, MAPK, PI3K, Janus kinase—STAT3 (JAK-STAT3), Wnt, HH, Notch, TGFβ, NFκB, JAK-STAT1/2 type I interferon, and JAK-STAT1/2 type II interferon. The development and validation of this technology for measuring pathway activity have been described in detail before [12,13,14,15,16,17]. For each pathway, the activity is calculated from mRNA levels of a defined set of high-evidence direct transcription factor target genes (20–30 per pathway) using a Bayesian network-based probabilistic computational model. Each model has been calibrated and validated using ‘ground truth’ samples, meaning samples with known both low and high STP activity. The calculated pathway activity score is normalized on a scale from 0–100. The set of target gene mRNA levels that serves as the input for the STP model can be measured by microarray (Affymetrix) and RNA sequencing. Activity of FOXO transcription factor is negatively regulated by the PI3K pathway, and FOXO activity can be used as an inverse readout for the PI3K pathway activity [13]. However, FOXO activity increases in the presence of oxidative stress, which complicates the interpretation of PI3K activity. For that reason, we also measured SOD2 levels to differentiate between increased FOXO activity because of decreased PI3K activity, or because of increased oxidative stress (Figure 2A) [13]. Two models have been developed for the JAK-STAT1/2 STP. The JAK-STAT1/2 type I interferon model measures JAK-STAT1/2 STP activity induced by type I interferons, and the STAT1/2 type II interferon model measures activity of the JAK-STAT1/2 STP induced by both type I and type II interferons [17].

### 4.3. Microarray Data Quality Control

Quality control (QC) was performed on Affymetrix (U133 Plus 2.0 Array) data of each sample from every dataset, as previously described [14]. QC parameters include the following: average value of all probe intensities, presence of negative or extremely high (>16-bit) intensity values, poly-A RNA (sample preparation spike-ins) and labelled cRNA (hybridization spike-ins) controls, GAPDH and ACTB 3′/5′ ratio, center of intensity, and values of positive and negative border controls. QC parameters were determined by AffyQCReport package in R, and an RNA degradation value was determined by the AffyRNAdeg function from the Affymetrix package in R (version 2024.12.1+563) [48]. Samples that failed QC were excluded from analysis. GSE15471 failed the Cmoff criterion for all samples. Cmoff calculates an equal distribution of the negative control samples on a chip. Nevertheless, GSE15471 was included in our analysis since STP activity was comparable with a similar dataset that passed QC (GSE16515) and because of the unique size of the dataset (36 paired samples).

### 4.4. Cell Typing of Samples in Datasets Using Deconvolution Software

Presence of other cell types in normal controls and tumor samples influences measured STP activity. Especially in PDAC tumors, this is relevant, since they are characterized by a relatively low number of epithelial tumor cells [42]. The presence of different cell types in the samples was identified using DCDC-Tx proprietary deconvolution software based on lineage-specific mRNA expression. The deconvolution software tool was validated on cell samples with known cellular content (supplements validation deconvolution software).

### 4.5. Statistical Analysis and Generation of Plots

Boxplots, individual sample, and correlation plots were generated using GraphPad Prism version 10.4.0 (GraphPad Software Incorporated, Boston, MA, USA, RRID:SCR_002798). Statistical analyses were performed with built-in tests in GraphPad Prism. Normality test was performed on GSE15471 and GSE16515, which showed non-Gaussian distribution. A similar distribution was assumed for the other datasets. Two-sided Mann–Whitney U-test was performed to compare unpaired samples (GSE32676, GSE71989, comparison between datasets), and Wilcoxon signed-rank test was performed to compare paired samples (GSE15471 and GSE16515). A *p*-value ≤ 0.01 was considered statistically significant [49].

## Figures and Tables

**Figure 1 ijms-26-11385-f001:**
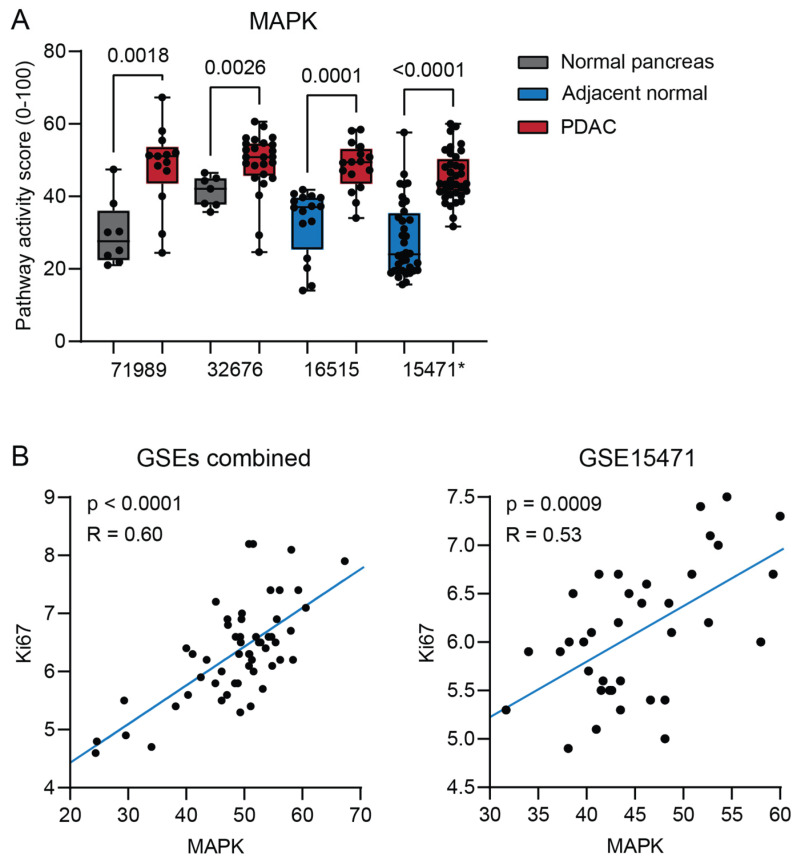
(**A**) MAPK signal transduction pathway (STP) activity of pancreatic ductal adenocarcinoma (PDAC) compared with normal pancreas (unpaired, GSE71989 and GSE32676) and normal adjacent pancreas (paired, GSE16515 and GSE15471). (**B**) Correlation of MAPK STP activity with Ki67 expression in PDAC tumor samples from three combined datasets (GSE71989, GSE32676, and GSE16515) and GSE15471. *p*-values ≤ 0.05 are depicted in numbers. * GSE15471 failed quality control, see Section 4.

**Figure 2 ijms-26-11385-f002:**
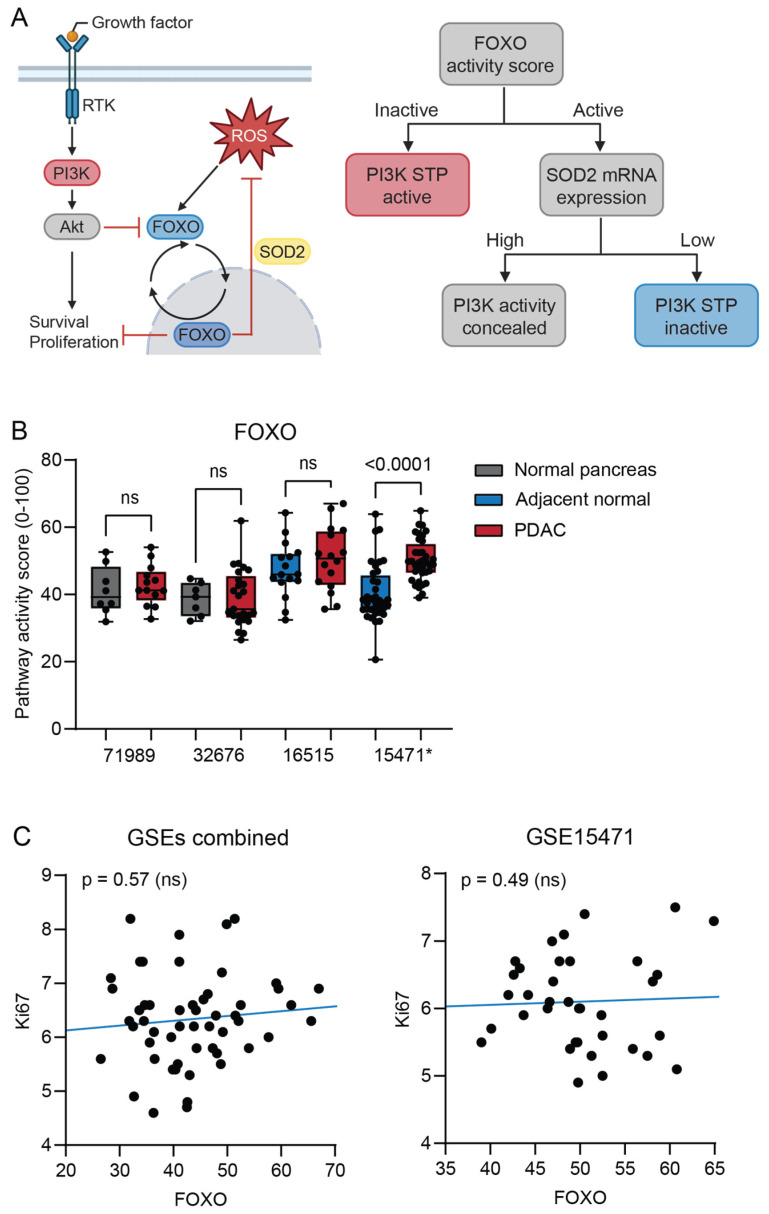
(**A**) Schematic interpretation of the PI3K pathway. When PI3K is activated, FOXO activity is blocked and FOXO translocates to the cytoplasm. Oxidative stress induces alternative activation of FOXO, with the function of protecting against reactive oxygen species (ROS). Schematic representation of how PI3K STP activity is determined from FOXO activity and SOD2 expression. (**B**) Signal transduction pathway (STP) activity of FOXO in pancreatic ductal adenocarcinoma (PDAC) compared with normal pancreas (unpaired, GSE71989 and GSE32676) and normal adjacent pancreas (paired, GSE16515 and GSE15471). (**C**) Correlation of FOXO STP activity with Ki67 expression of individual PDAC tumor samples from three combined datasets (GSE71989, GSE32676, and GSE16515) and GSE15471. *p*-value > 0.01 is considered non-significant (ns). *p*-values ≤ 0.05 are depicted in numbers. * GSE15471 failed quality control.

**Figure 3 ijms-26-11385-f003:**
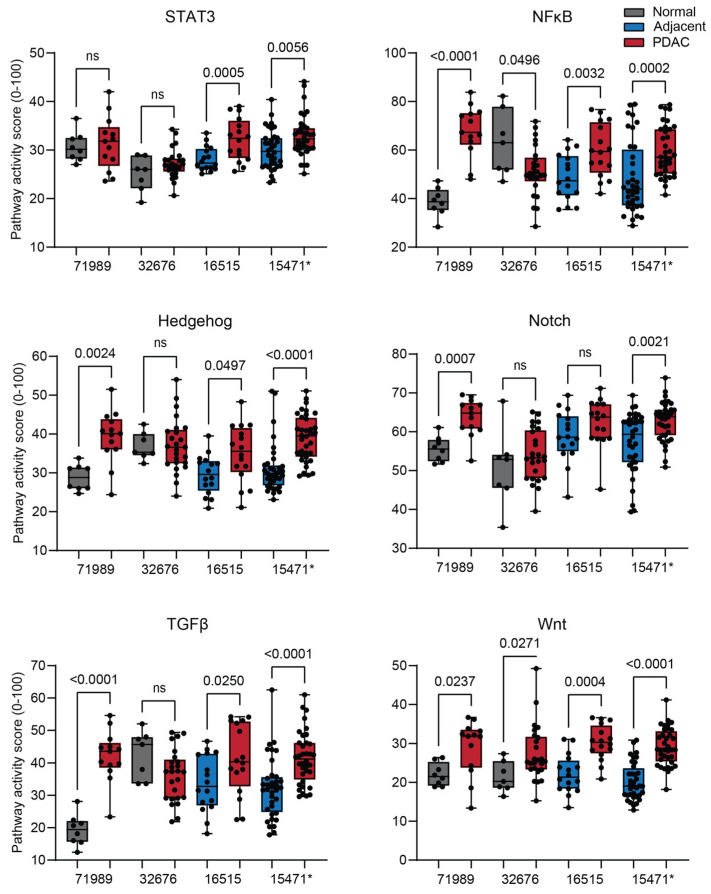
Signal transduction pathway (STP) activity scores of the other STPs altered in pancreatic ductal adenocarcinoma (PDAC). PDAC is compared with normal pancreas (unpaired, GSE71989 and GSE32676) and normal adjacent pancreas (paired, GSE16515 and GSE15471). STPs showing significant alteration in two or more datasets are depicted. *p*-value > 0.01 is considered non-significant (ns). *p*-values ≤ 0.05 are depicted in numbers. * GSE15471 failed quality control.

**Figure 4 ijms-26-11385-f004:**
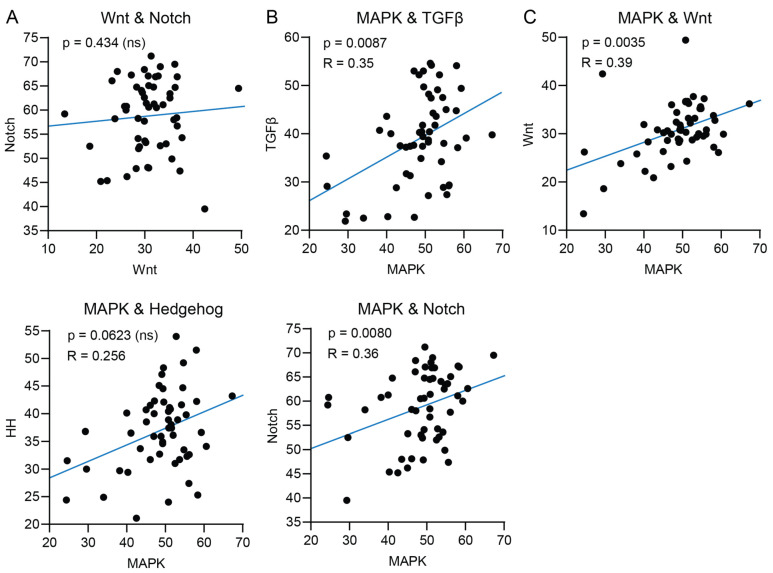
STP activity scores from PDAC tumor samples of GSE16515, GSE32676, and GSE71989 were pooled. (**A**) Correlation of Wnt signal transduction pathway (STP) activity with Notch STP activity of individual tumor samples. (**B**) Correlation of MAPK signal transduction pathway (STP) activity in tumor samples with TGFβ and (**C**) Wnt, Notch, and Hedgehog (HH) STP activity.

**Figure 5 ijms-26-11385-f005:**
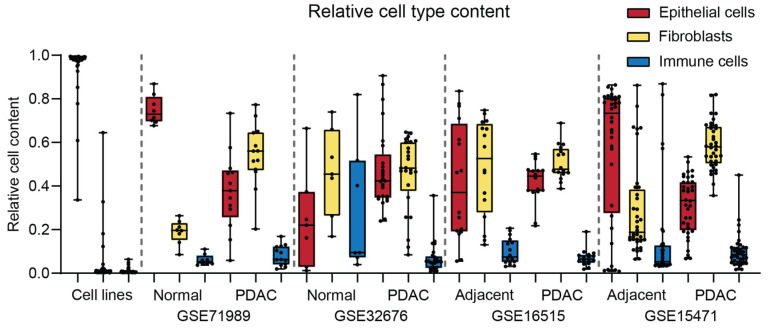
Relative cell type content (total = 1.0) determined using deconvolution software for cell lines, normal pancreas, normal tumor-adjacent pancreas, and pancreatic ductal adenocarcinoma (PDAC).

**Figure 6 ijms-26-11385-f006:**
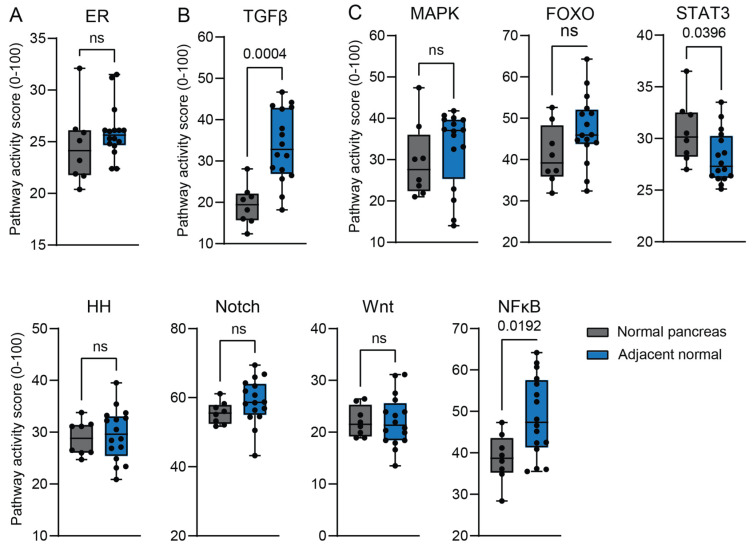
Comparison of signal transduction pathway (STP) activity scores of normal (healthy donor) pancreas (GSE71989) with tumor-adjacent normal pancreas (GSE16515). (**A**) Estrogen receptor (ER) pathway activity is shown as a control, since its activity is not altered in PDAC. (**B**) TGFβ STP activity is increased in tumor-adjacent tissue. (**C**) Pathways with no differences in STP activity. A *p*-value > 0.01 is considered non-significant (ns). Values are depicted in numbers if *p* ≤ 0.05.

**Figure 7 ijms-26-11385-f007:**
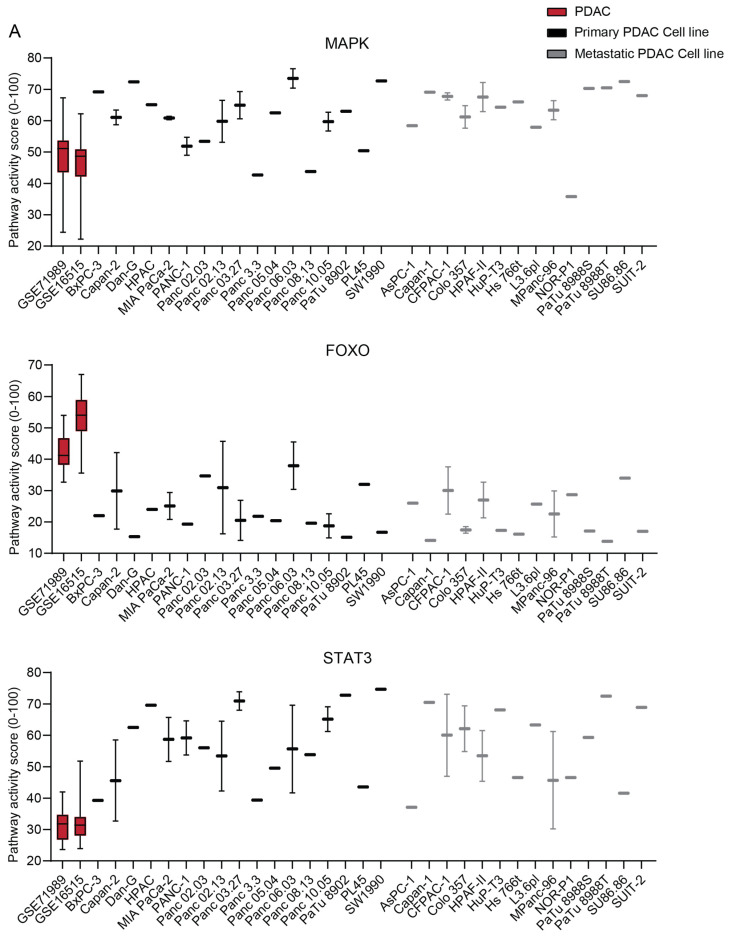
(**A**) Signal transduction pathway (STP) activity scores of pancreatic ductal adenocarcinoma (PDAC) tumor samples (GSE71989 and GSE16515) and cell lines derived from primary and metastatic PDAC. (**B**) Ki67 expression scores of PDAC tumor samples and cell lines. Cell line STP activity scores are combined to determine differences with patient samples of GSE16515.

**Table 1 ijms-26-11385-t001:** Paired MAPK and TGFβ STP scores.

	MAPK		TGFβ	
Sample	Adjacent	Tumor	Adjacent	Tumor
1	40.1	58.4	31.5	37.1
2	15.3	58.1	34.1	54.1
3	32.5	54.8	26.5	38.0
4	40.7	53.7	46.7	52.2
5	38.6	51.6	37.9	54.2
6	20.2	50.8	25.7	38.3
7	37.3	49.6	21.3	49.7
8	36.9	49.5	31.3	53.0
9	22.9	49.3	44.2	41.8
10	39.7	47.2	43.4	22.7
11	39.6	47.1	42.6	53.0
12	35.9	46.1	28.4	31.3
13	33.1	42.5	36.4	28.8
14	41.8	41.1	43.0	40.0
15	14.0	38.2	27.9	40.7
16	37.2	34.0	18.2	22.5

MAPK and TGFβ signal transduction pathway (STP) activity scores of the individual paired samples from GSE16515: adjacent normal pancreas and PDAC tumor. Samples are ranked by tumor MAPK STP activity. STP activity scores are displayed with colors, showing the highest activity in red, medium in white, and the lowest in blue.

**Table 2 ijms-26-11385-t002:** GEO datasets included in the study.

GEO Dataset	Samples	Number of Samples	Control	Selection
GSE15471 [19]	Tumor Control	36 36	Tumor-adjacent	Microscopic
GSE16515 [20]	Tumor Control	16 16	Tumor-adjacent	Microscopic
GSE32676 [21]	Tumor Control	25 7	Normal pancreas	Microscopic >30% tumor cells
GSE71989 [22]	Tumor Control	13 8	Normal pancreas	Not specified
GSE17891 [34]	Tumor Cell lines	27 20	None	Laser microdissection
GSE21654 [35]	Cell lines	22		

Tumor samples from GSE17891 were excluded as they lacked normal tissue controls.

## Data Availability

The data generated in this study are available within the article and its Appendix A. Further inquiries can be directed to the corresponding author. Expression profile data analyzed in this study were obtained from the GEO database at GSE15471, GSE16515, GSE32676, GSE71989, GSE17891 and GSE21654 and are available in the GEO database repository, https://www.ncbi.nlm.nih.gov/geo/ (last accessed on 12 May 2025).

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
