# Peer review of "Quantification of Signal Transduction Pathway Activity in Pancreatic Ductal Adenocarcinoma"

_ijms, 2025, doi:10.3390/ijms262311385_

Round 1
Reviewer 1 Report
Comments and Suggestions for Authors
This is a well-written and methodologically sound study that applies STAP-STP to analyze signal transduction pathway activity in PDAC. The work is clearly presented, and provides valuable biological insights with strong translational implications. The analytical framework is very good, incorporating multiple GEO datasets and appropriate quality controls, and the results are biologically consistent with known PDAC signaling.
While the study relies on public datasets and lacks experimental validation, the authors acknowledge these limitations appropriately. Overall, this is a robust computational analysis that contributes meaningfully to the understanding of PDAC signaling and therapeutic targeting.
The major strengths of the study are it addresses a clinically important challenge and applies a novel analytical framework that goes beyond standard gene expression approaches.
Additionally, integration of multiple independent GEO datasets adds robustness and reproducibility. The findings are biologically consistent with known PDAC signaling such as MAPK, STAT3 etc. Moreover the analysis of the PI3K–FOXO axis reflects thoughtful interpretation of complex biological interactions.
Despite the positive aspects I mentioned here, I have a few concerns and the following points should be addressed.
- The authors should explicitly acknowledge that the study is based on publicly available datasets and lacks experimental validation.
- It is important to clarify that pathway activation patterns are associative rather than causal, and present speculative interpretations as hypotheses.
- It would be better to add a brief discussion on how clinical correlations (such as stage or outcome) could further enhance the translational impact.
- I have a problem with the figures. Simplify or reorganize multi-panel figures to improve visual clarity.
- Additionally, authors may include a short paragraph summarizing key limitations such as use of retrospective data, computational inference, and lack of in vitro confirmation.
- To further help the general audience, please provide a concise explanation and may be a schematic of how STAP-STP calculates pathway scores. This will really help readers who are unfamiliar with the method.
- I think it is important to emphasize that while many identified pathways are known in PDAC, this work’s innovation lies in quantifying and comparing their activity using a validated systems approach.
Overall, this work represents a valuable contribution to the field and is suitable for publication upon revising it as suggested.
Author Response
We would like to thank you for taking the time to review our manuscript and the useful comments and suggestions. Please find the detailed point-by-point responses below with the corresponding revisions/corrections highlighted in blue in the resubmitted files. We hope you will be satisfied by our responses and revisions. We have summarized all changes to the manuscript at the end of this document.
Reviewer 1:
This is a well-written and methodologically sound study that applies STAP-STP to analyze signal transduction pathway activity in PDAC. The work is clearly presented, and provides valuable biological insights with strong translational implications. The analytical framework is very good, incorporating multiple GEO datasets and appropriate quality controls, and the results are biologically consistent with known PDAC signaling.
While the study relies on public datasets and lacks experimental validation, the authors acknowledge these limitations appropriately. Overall, this is a robust computational analysis that contributes meaningfully to the understanding of PDAC signaling and therapeutic targeting.
The major strengths of the study are it addresses a clinically important challenge and applies a novel analytical framework that goes beyond standard gene expression approaches.
Additionally, integration of multiple independent GEO datasets adds robustness and reproducibility. The findings are biologically consistent with known PDAC signaling such as MAPK, STAT3 etc. Moreover the analysis of the PI3K–FOXO axis reflects thoughtful interpretation of complex biological interactions.
Despite the positive aspects I mentioned here, I have a few concerns and the following points should be addressed.
- The authors should explicitly acknowledge that the study is based on publicly available datasets and lacks experimental validation.
We thank the reviewer for this comments, it is indeed important to emphasize that the datasets were obtained from the GEO database and we did not perform our own experiments as validation. In lines 27 and 69 we added the following: publicly available
A comment on this limitation has been added to lines 431-434 in the discussion: Our study relies on these previously generated datasets, which likely differ in methodology and quality and STP activity should be confirmed in future experiments when testing drugs interfering on these pathways.
And lines 460-462: STP activity of cell lines should be verified with consistent culturing conditions and the influence of serum concentration in culture medium should be evaluated.
- It is important to clarify that pathway activation patterns are associative rather than causal, and present speculative interpretations as hypotheses.
We thank the reviewer for this important comment. Alterations in pathway activity will results in alterations in cellular function, but may be a consequence of for example a driver mutation instead of causal. The MAPK pathway is probably a driver pathway due to >90% of PDAC patients having activating KRAS mutations, but we do not prove that in this study. Interference on STPs can provide further evidence on which STPs drive PDAC development and proliferation as a consequence of activating mutations and which STPs are altered as a consequence of the disease having developed. We have added the text to adjust statements and add a section explaining this:
Lines 72-73: Alterations in STP activity may drive oncogenesis play a role in tumor growth and metastasis and pose opportunities for therapeutic interventions.
Lines 414-416: High activity of the developmental pathway TGFβ may play a causal role in causing epithelial-mesenchymal transition (EMT) due to the high prevalence of SMAD gene mutations in PDAC [18,37].
References 37 was added here as extra reference on SMAD gene mutations and TGFβ in PDAC: Dardare, J.; Witz, A.; Merlin, J.L.; Gilson, P.; Harle, A. SMAD4 and the TGFbeta Pathway in Patients with Pancreatic Ductal Adenocarcinoma. Int J Mol Sci 2020, 21, doi:10.3390/ijms21103534.
Lines 463-469: STAP-STP analysis provides unique insights into the abnormal functioning of STP in PDAC pathways that drive oncogenesis and disease progression of PDAC. MAPK and PI3K are probably STPs that drive tumor growth given that more than 90% of PDAC tumors have KRAS mutations. TGFβ may be another important pathway, given the high mutation rate in SMAD genes. Interfering on specific STP and quantifying the influence on all STPs can provide evidence on which STPs drive tumor growth and how these pathways crosstalk.
- It would be better to add a brief discussion on how clinical correlations (such as stage or outcome) could further enhance the translational impact.
Thank you for this comments, we have added the following to lines 469-472 in the discussion: Correlating STP activity to clinical outcomes, such as disease-free interval and overall survival, increases the probability that these pathways are involved in proliferation and strengthens the rationale to interfere in these pathways.
- I have a problem with the figures. Simplify or reorganize multi-panel figures to improve visual clarity.
We hope that we have made major improvements to figures 1-6 by editing the width of the box plots, simplifying some figures and removal of unnecessary information. We saw no possibility to make Figure 7 into a single panel figure, as it would become too small to maintain readability.
- Additionally, authors may include a short paragraph summarizing key limitations such as use of retrospective data, computational inference, and lack of in vitro confirmation.
The aim of the study was to explore STP activity for PDAC and relied on previously generated datasets. Quality and methodology could differ and STP activity may be validated in future experiments. STP activity can be confirmed when testing drugs interfering on these pathways. The following was added to lines 431-434 in the discussion:
Our study relies on these previously generated datasets, which likely differ in methodology and quality and should be confirmed in future experiments when testing drugs interfering on these pathways.
- To further help the general audience, please provide a concise explanation and may be a schematic of how STAP-STP calculates pathway scores. This will really help readers who are unfamiliar with the method.
We have added a graphical abstract that shows how activity scores are calculated from targets genes of transcription factors. We hope this would clarify our methodology for the reader.
- I think it is important to emphasize that while many identified pathways are known in PDAC, this work’s innovation lies in quantifying and comparing their activity using a validated systems approach.
We think this is a great addition and added the following to lines 36-37 in the abstract: This is the first time that STP activity has been quantified in PDAC.
And the following to lines 474-476 in the discussion: This is the first time STP activity has been quantified for PDAC, as this Activity of STPs cannot be performed quantified with conventional bioinformatic analysis methods, as has been described before [9].
Overall, this work represents a valuable contribution to the field and is suitable for publication upon revising it as suggested.
Summary of revisions
Title: Quantification of Ssignal transduction pathway activity analysis in pancreatic ductal adenocarcinoma in search for new therapeutic targets
Addition of graphical abstract
Abstract:
Line 27: publicly available
Lines 36-37: This is the first time that STP activity has been quantified in PDAC.
Keywords:
We have removed ’pancreatic cancer’ as keyword, as pancreatic ductal adenocarcinoma has already been mentioned. STAP was written out as simultaneous transcriptome-based activation profiling and added as keyword. In addition, ‘MAPK’ was added as important STP in PDAC.
Keywords were reordered to mention: STPs, disease, technique, therapy.
Introduction:
Line 69: publicly available
Lines 72-73: Alterations in STP activity may drive oncogenesis play a role in tumor growth and metastasis and pose opportunities for therapeutic interventions.
Results:
Edited Figures 1-6
Line 117 added: (Figure 2A)
Line 119 changed to: Figure 2BA
Line 122 changed to: (Figure 2CB).
Lines 149-152: Figure 2. (A) Schematic interpretation of the PI3K pathway. When PI3K is activated, FOXO activity is blocked and FOXO translocates to the cytoplasm. Oxidative stress induces alternative activation of FOXO with the function to protect against reactive oxygen species (ROS). Schematic representation on how PI3K STP activity is determined from FOXO activity and SOD2 expression.
Line 201 added: altered
Discussion:
Line 414: (ref 41).
Lines 414-416: High activity of the developmental pathway TGFβ may play a causal role in causing epithelial-mesenchymal transition (EMT) due to the high prevalence of SMAD gene mutations in PDAC [18,37].
Lines 423 added: (Figure 2A)
Lines 431-434: Our study relies on these previously generated datasets, which likely differ in methodology and quality and STP activity should be confirmed in future experiments when testing drugs interfering on these pathways.
Lines 440-442: A limitation of the deconvolution software is that it is based RNA expression, which may differ from protein expression.
Lines 460-462: STP activity of cell lines should be verified with consistent culturing conditions and the influence of serum concentration in culture medium should be evaluated.
Lines 463-472 in the discussion: STAP-STP analysis provides unique insights into the abnormal functioning of signal transduction pathways in PDAC. pathways that drive oncogenesis and disease progression of PDAC. MAPK and PI3K are probably STPs that drive tumor growth given that more than 90% of PDAC tumors have KRAS mutations. TGFβ may be another important pathway, given the high mutation rate in SMAD genes [37]. Interfering on specific STPs and quantifying the influence on all STPs provides evidence which STPs drive tumor growth and how these pathways crosstalk. Correlating STP activity to clinical outcomes, such as disease-free interval and overall survival, increases the probability that these pathways are involved in proliferation and strengthens the rationale to interfere in these pathways.
Lines 474-476: This is the first time STP activity has been quantified for PDAC, as this Activity of STPs cannot be performed quantified with conventional bioinformatic analysis methods, as has been described before [9]
Line 481 added: aberrant
Lines 486-489: STAP-STP technology is unique in monitoring the influence of drugs on all STPs and identify crosstalk resulting in therapy resistance, and helping to develop combinational therapy. It also STAP-STP technology uniquely enables development and evaluation of cancer differentiation therapy by affecting all relevant pathways.
Methods:
Updated the references in Table 2
Lines 530-531: Activity of Forkhead box O (FOXO) transcription factor
Lines 534-535 :superoxide dismutase 2 (SOD2)
Line 536 added: (Figure 2A)
References:
Checked and updated references
Reference 37 was added: Dardare, J.; Witz, A.; Merlin, J.L.; Gilson, P.; Harle, A. SMAD4 and the TGFbeta Pathway in Patients with Pancreatic Ductal Adenocarcinoma. Int J Mol Sci 2020, 21, doi:10.3390/ijms21103534.

Reviewer 2 Report
Comments and Suggestions for Authors
The authors identify abnormal STP activity in patients with pancreatic ductal adenocarcinoma using simultaneous transcriptome-based activation profiling (STAP)-STP technology. The results seem to be good but the manuscript was not well-written. Comments:
- The introduction should include more information such as the previous methods and works about this research. The background and purpose of this study should be carefully discussed in Introduction.
- The Abstract should be modified and refined. For example, the importance and conclusion of this work should be mentioned in Abstract.
- The keywords should be revised and some of them could be removed. The full names of keywords should be provided.
- The English could be improved to more clearly express the research. For example, it is difficult for the reviewer to understand the title “Signal transduction pathway analysis in pancreatic ductal adenocarcinoma in search for new therapeutic targets”.
- The results should be integrated with the discussions so that the readers can understand this work easily. The results should be discussed but not just displayed.
- The table and reference format should be revised.
- No conclusion section was found in the manuscript.
Author Response
We would like to thank you for taking the time to review our manuscript and the useful comments and suggestions. Please find the detailed point-by-point responses below with the corresponding revisions/corrections highlighted in blue in the resubmitted files. We hope you will be satisfied by our responses and revisions. We have summarized all changes to the manuscript at the end of this document.
Reviewer 2:
The authors identify abnormal STP activity in patients with pancreatic ductal adenocarcinoma using simultaneous transcriptome-based activation profiling (STAP)-STP technology. The results seem to be good but the manuscript was not well-written. Comments:
- The introduction should include more information such as the previous methods and works about this research. The background and purpose of this study should be carefully discussed in Introduction.
We thank the reviewer for their comments. We have referred to development and validation of the STAP-STP per pathway in the introduction: references 12-17. To further explain how STP scores are calculated, we have added a graphical abstract that summarizes the methods and altered STPs.
The innovation and purpose of this study lies in identifying which STPs are aberrantly active in PDAC and quantification of STP activity. This forms the basis for drug development as drugs targeting these aberrantly expressed STPs can be used and its effect on all STPs can be quantified. We have tried to better explain this:
Lines 474-476: The abnormally active STPs are in line with previously described pathways that play a role in PDAC [7,8,11]. This is the first time STP activity has been quantified for PDAC, as this Activity of STPs cannot be performed quantified with conventional bioinformatic analysis methods, as has been described before [9]
Lines 486-489 in the discussion: STAP-STP technology is unique in monitoring the influence of drugs on all STPs and identify crosstalk resulting in therapy resistance, and helping to develop combinational therapy. It also STAP-STP technology uniquely enables development and evaluation of cancer differentiation therapy by affecting all relevant pathways.
- The Abstract should be modified and refined. For example, the importance and conclusion of this work should be mentioned in Abstract.
We thank the reviewer for this comment and have therefore added the following statement in lines 36-37: This is the first time that STP activity has been quantified in PDAC. To emphasize the uniqueness of the data, which is followed by a conclusion how this can be used to develop and evaluated drugs targeting aberrantly active STPs in PDAC.
- The keywords should be revised and some of them could be removed. The full names of keywords should be provided.
We thank the reviewer for noticing this. We have removed ’pancreatic cancer’ as keyword, as pancreatic ductal adenocarcinoma has already been mentioned. STAP was written out as simultaneous transcriptome-based activation profiling and added as keyword. In addition, important STP keywords based on the aberrantly active pathways in the study were added: ‘MAPK, Wnt, Hedgehog, Notch, TGFβ, NFκB’
Keywords were reordered to mention: STPs, disease, technique, therapy.
- The English could be improved to more clearly express the research. For example, it is difficult for the reviewer to understand the title “Signal transduction pathway analysis in pancreatic ductal adenocarcinoma in search for new therapeutic targets”.
Thank you for this observations as the title is essential to summarize the study in one sentence. We have changed the title to: ‘Quantification of signal transduction pathway activity in pancreatic ductal adenocarcinoma’
This better shows what was done in this study, quantification of STP activity, and shows its uniqueness. We removed ‘therapeutic targets’ as no therapies are tested in this study, although the next step would be interfering on aberrantly expressed STPs and quantifying its influence on all STPs.
We have checked the full text for English and adjusted text where necessary.
- The results should be integrated with the discussions so that the readers can understand this work easily. The results should be discussed but not just displayed.
We thank the reviewer for this consideration, but we would prefer to keep the result and discussion section separated as it is. We already added some extra explanation to the results section by explaining the prevalence of KRAS mutations and its role in activation of the MAPK and PI3K pathways before showing the STP activity of these pathways. In addition, we made correlations between STPs in Figure 4 and discuss in the results that these correlation were made based on literature.
Furthermore, we hope that the graphical abstract and addition of Figure 2A to Figure 2 will help the reader to better understand the contents of this study.
The following has been added to the legend to describe Figure 2A in lines 149-152: Figure 2. (A) Schematic interpretation of the PI3K pathway. When PI3K is activated, FOXO activity is blocked and FOXO translocates to the cytoplasm. Oxidative stress induces alternative activation of FOXO with the function to protect against reactive oxygen species (ROS). Schematic representation on how PI3K STP activity is determined from FOXO activity and SOD2 expression.
- The table and reference format should be revised.
We thank the reviewer for this observation. We have checked and updated the references in MDPI/IJMS format.
A mistake with a reference in the text has been removed in line 414: (ref 41).
We have also updated the references in Table 2.
- No conclusion section was found in the manuscript.
We thank the reviewer for noticing this. It is true that there is no conclusion header in this manuscript, although a conclusion is present in the last paragraph of the discussion. The IJMS guidelines state: Conclusions: This section is not mandatory but can be added to the manuscript if the discussion is unusually long or complex.
We saw no additional value in adding a separate conclusion, since we consider the discussion not to be of such lengths and complexity. A separate conclusion section can still be added if desired by the reviewer.
Summary of revisions
Title: Quantification of Ssignal transduction pathway activity analysis in pancreatic ductal adenocarcinoma in search for new therapeutic targets
Addition of graphical abstract
Abstract:
Line 27: publicly available
Lines 36-37: This is the first time that STP activity has been quantified in PDAC.
Keywords:
We have removed ’pancreatic cancer’ as keyword, as pancreatic ductal adenocarcinoma has already been mentioned. STAP was written out as simultaneous transcriptome-based activation profiling and added as keyword. In addition, ‘MAPK’ was added as important STP in PDAC.
Keywords were reordered to mention: STPs, disease, technique, therapy.
Introduction:
Line 69: publicly available
Lines 72-73: Alterations in STP activity may drive oncogenesis play a role in tumor growth and metastasis and pose opportunities for therapeutic interventions.
Results:
Edited Figures 1-6
Line 117 added: (Figure 2A)
Line 119 changed to: Figure 2BA
Line 122 changed to: (Figure 2CB).
Lines 149-152: Figure 2. (A) Schematic interpretation of the PI3K pathway. When PI3K is activated, FOXO activity is blocked and FOXO translocates to the cytoplasm. Oxidative stress induces alternative activation of FOXO with the function to protect against reactive oxygen species (ROS). Schematic representation on how PI3K STP activity is determined from FOXO activity and SOD2 expression.
Line 201 added: altered
Discussion:
Line 414: (ref 41).
Lines 414-416: High activity of the developmental pathway TGFβ may play a causal role in causing epithelial-mesenchymal transition (EMT) due to the high prevalence of SMAD gene mutations in PDAC [18,37].
Lines 423 added: (Figure 2A)
Lines 431-434: Our study relies on these previously generated datasets, which likely differ in methodology and quality and STP activity should be confirmed in future experiments when testing drugs interfering on these pathways.
Lines 440-442: A limitation of the deconvolution software is that it is based RNA expression, which may differ from protein expression.
Lines 460-462: STP activity of cell lines should be verified with consistent culturing conditions and the influence of serum concentration in culture medium should be evaluated.
Lines 463-472 in the discussion: STAP-STP analysis provides unique insights into the abnormal functioning of signal transduction pathways in PDAC. pathways that drive oncogenesis and disease progression of PDAC. MAPK and PI3K are probably STPs that drive tumor growth given that more than 90% of PDAC tumors have KRAS mutations. TGFβ may be another important pathway, given the high mutation rate in SMAD genes [37]. Interfering on specific STPs and quantifying the influence on all STPs provides evidence which STPs drive tumor growth and how these pathways crosstalk. Correlating STP activity to clinical outcomes, such as disease-free interval and overall survival, increases the probability that these pathways are involved in proliferation and strengthens the rationale to interfere in these pathways.
Lines 474-476: This is the first time STP activity has been quantified for PDAC, as this Activity of STPs cannot be performed quantified with conventional bioinformatic analysis methods, as has been described before [9]
Line 481 added: aberrant
Lines 486-489: STAP-STP technology is unique in monitoring the influence of drugs on all STPs and identify crosstalk resulting in therapy resistance, and helping to develop combinational therapy. It also STAP-STP technology uniquely enables development and evaluation of cancer differentiation therapy by affecting all relevant pathways.
Methods:
Updated the references in Table 2
Lines 530-531: Activity of Forkhead box O (FOXO) transcription factor
Lines 534-535 :superoxide dismutase 2 (SOD2)
Line 536 added: (Figure 2A)
References:
Checked and updated references
Reference 37 was added: Dardare, J.; Witz, A.; Merlin, J.L.; Gilson, P.; Harle, A. SMAD4 and the TGFbeta Pathway in Patients with Pancreatic Ductal Adenocarcinoma. Int J Mol Sci 2020, 21, doi:10.3390/ijms21103534.

Reviewer 3 Report
Comments and Suggestions for Authors
This manuscript presents an in-depth bioinformatic analysis of signal transduction pathway (STP) activity in pancreatic ductal adenocarcinoma (PDAC) using STAP-STP (simultaneous transcriptome-based activation profiling) technology. The study identifies the aberrant activation of several key pathways (MAPK, STAT3, Wnt, Hedgehog, Notch, TGFβ, NFκB) and explores their interrelationships and implications for therapeutic targeting.
The topic is timely, the rationale is well-founded, and the methodology is clearly explained. The manuscript is well organised and written in correct scientific English. Overall, it represents a valuable contribution to the understanding of molecular signalling in PDAC and the development of potential pathway-based therapeutic strategies.
However, the study relies exclusively on transcriptomic data from public GEO datasets. While this ensures breadth, it limits biological validation. Please discuss more explicitly how future experimental work (e.g., proteomic validation, phospho-signalling assays, or functional inhibition studies) could corroborate the activities of the predicted pathways.
Furthermore, the section on FOXO and oxidative stress (Figure 2 and related text) is conceptually complex. The rationale linking oxidative stress to FOXO activation is sound, but the explanation could be streamlined. Consider adding a concise schematic to illustrate how FOXO activity confounds PI3K interpretation in PDAC.
Then the discussion should elaborate further on how the identified pathway activity profiles could guide combination therapies or differentiation therapy approaches in PDAC. For instance, what is the rationale for targeting MAPK and TGFβ together, or how might STAP-STP assist in drug screening?
A short paragraph outlining study limitations (dataset heterogeneity, lack of experimental confirmation, potential biases in deconvolution) would strengthen the manuscript and show critical awareness of the methodology.
-Figures 3–4 are information-dense and could be made clearer with simplified color coding and enlarged axis labels.
-A summary schematic illustrating the key activated pathways and their interactions (MAPK, STAT3, Wnt, Notch, TGFβ, NFκB) in PDAC would help readers visualize the main findings.
-Ensure consistent formatting for gene names (e.g., KRAS, FOXO, SOD2).
-The GEO accession numbers are clearly listed; however, indicate whether any processed data or STP activity matrices will be made available as supplementary material.
Author Response
We would like to thank you for taking the time to review our manuscript and the useful comments and suggestions. Please find the detailed point-by-point responses below with the corresponding revisions/corrections highlighted in blue in the resubmitted files. We hope you will be satisfied by our responses and revisions. We have summarized all changes to the manuscript at the end of this document.
Reviewer 3:
This manuscript presents an in-depth bioinformatic analysis of signal transduction pathway (STP) activity in pancreatic ductal adenocarcinoma (PDAC) using STAP-STP (simultaneous transcriptome-based activation profiling) technology. The study identifies the aberrant activation of several key pathways (MAPK, STAT3, Wnt, Hedgehog, Notch, TGFβ, NFκB) and explores their interrelationships and implications for therapeutic targeting.
The topic is timely, the rationale is well-founded, and the methodology is clearly explained. The manuscript is well organised and written in correct scientific English. Overall, it represents a valuable contribution to the understanding of molecular signalling in PDAC and the development of potential pathway-based therapeutic strategies.
- However, the study relies exclusively on transcriptomic data from public GEO datasets. While this ensures breadth, it limits biological validation. Please discuss more explicitly how future experimental work (e.g., proteomic validation, phospho-signalling assays, or functional inhibition studies) could corroborate the activities of the predicted pathways.
We thank the reviewer for these important considerations.
STAP-STP technology has been developed and extensively validated for each described STP in previous studies: refs 12-17. Pathway activity scores have been determined based on ground truth samples where activity is fully suppressed or activated. Proteomic validation is not applicable, but phospho-signaling assays and inhibition studies could be performed. Although not necessary for validation of pathways activity, as this has been extensively researched before, especially inhibition studies could be performed to determine which pathways drive PDAC, as our aberrant STP activity scores are only associative. Interfering on STPs and quantifying its effects could help to show which pathways crosstalk and drive proliferation in PDAC. The following was added to lines 463-472 in the discussion: STAP-STP analysis provides unique insights into the abnormal functioning of signal transduction pathways in PDAC. pathways that drive oncogenesis and disease progression of PDAC. MAPK and PI3K are probably STPs that drive tumor growth given that more than 90% of PDAC tumors have KRAS mutations. TGFβ may be another important pathway, given the high mutation rate in SMAD genes [37]. Interfering on specific STPs and quantifying the influence on all STPs provides evidence which STPs drive tumor growth and how these pathways crosstalk. Correlating STP activity to clinical outcomes, such as disease-free interval and overall survival, increases the probability that these pathways are involved in proliferation and strengthens the rationale to interfere in these pathways.
- Furthermore, the section on FOXO and oxidative stress (Figure 2 and related text) is conceptually complex. The rationale linking oxidative stress to FOXO activation is sound, but the explanation could be streamlined. Consider adding a concise schematic to illustrate how FOXO activity confounds PI3K interpretation in PDAC.
We agree with the reviewer that the interpretation of the PI3K pathway from FOXO is complex. Therefore, we have added a schematic and graphical description of the PI3K pathway and interpretation of its activity via FOXO transcription factor activity to Figure 2.
The following has been added to the legend to describe Figure 2A in lines 149-152: Figure 2. (A) Schematic interpretation of the PI3K pathway. When PI3K is activated, FOXO activity is blocked and FOXO translocates to the cytoplasm. Oxidative stress induces alternative activation of FOXO with the function to protect against reactive oxygen species (ROS). Schematic representation on how PI3K STP activity is determined from FOXO activity and SOD2 expression.
- Then the discussion should elaborate further on how the identified pathway activity profiles could guide combination therapies or differentiation therapy approaches in PDAC. For instance, what is the rationale for targeting MAPK and TGFβ together, or how might STAP-STP assist in drug screening?
We thank the reviewer for this comment, as it would be the next step to use these aberrant STP activities for drug screening. We first need to determine which STPs drive tumor proliferation, as this is not tested in this study. Based on literature and mutations (KRAS and SMAD4), MAPK, PI3K and TGFβ are probably these pathways. Interfering on STPs and evaluation of all pathways with STAP-STP technology can help to determine which pathways drive proliferation in PDAC. Subsequent targeting of these pathways and crosstalk can be evaluated with STAP-STP technology. This has already been added to the discussion with question 1: lines 463-472.
When the driving pathways have been identified, interference on these pathways and evaluation of its effect on all STP with STAP-STP technology helps to identify crosstalk and therefore the development of combinational therapies. Therefore, the following has been added to lines 486-489 in the discussion: STAP-STP technology is unique in monitoring the influence of drugs on all STPs and identify crosstalk resulting in therapy resistance, and helping to develop combinational therapy. It also STAP-STP technology uniquely enables development and evaluation of cancer differentiation therapy by affecting all relevant pathways.
- A short paragraph outlining study limitations (dataset heterogeneity, lack of experimental confirmation, potential biases in deconvolution) would strengthen the manuscript and show critical awareness of the methodology.
For the patients samples, differences in methodology such between tissue selection and dissociation and RNA isolation can influence outcomes. Although a quality control is performed with our analysis, admixture of other cell types can also influence the results. The following was added to lines 431-434: Our study relies on these previously generated datasets, which likely differ in methodology and quality and STP activity should be confirmed in future experiments when testing drugs interfering on these pathways.
We added a statement about the deconvolution software, which is only based on RNA expression and not protein expression, to lines 440-442: A limitation of the deconvolution software is that it is based RNA expression, which may differ from protein expression.
The cell line data can be influenced by culture conditions, such a pH levels, percentage of serum added to culture medium and variations between batches of serum. Activity did indeed sometimes differ between two cell line measurements and the activity scores should be validated in separate cell line experiments with equal and optimized culture conditions. The following was added to lines 460-462 in the discussion: STP activity of cell lines should be verified with consistent culturing conditions and the influence of serum concentration in culture medium should be evaluated.
- Figures 3–4 are information-dense and could be made clearer with simplified color coding and enlarged axis labels.
We thank the reviewer of this observation and hope that we have made major improvements to figures 1-6.
For figure 4, we added titles to the correlation plots, so the reader can easier spot which pathways are being correlated instead of relying on the axis labels.
- A summary schematic illustrating the key activated pathways and their interactions (MAPK, STAT3, Wnt, Notch, TGFβ, NFκB) in PDAC would help readers visualize the main findings.
We thank the reviewer for this comments. It is important to clarify that the interaction between STPs still needs to be proven. This study provides an overview of the pathways that are probably involved and provides a basis to explore which pathways interact by interfering on these pathways and measuring the influence on activity score of all other pathways. In this study we do not prove any interactions between pathways yet. We have already added this to the discussion in lines 465-469 with your first comment.
The graphical abstract and addition of Figure 2A do help to visualize the components involved in a STP and summarizes the abnormally active pathways in PDAC.
- Ensure consistent formatting for gene names (e.g., KRAS, FOXO, SOD2).
We checked all gene name formatting and adjusted for consistency.
Lines 530-531 were edited as FOXO had already been written out before: Activity of Forkhead box O (FOXO) transcription factor
Same for SOD2 in lines 534-535:superoxide dismutase 2 (SOD2)
Due to the frequency and common use of KRAS and FOXO as abbreviations, we chose to only write out SOD2 in the figure legends (in this case in the supplementary figures depicting SOD2 expression levels)
- The GEO accession numbers are clearly listed; however, indicate whether any processed data or STP activity matrices will be made available as supplementary material.
The output excel matrices with pathways activity scores will only be available on request and not added as standard supplement. This is stated in the data availability statement (lines 598-599).
Summary of revisions
Title: Quantification of Ssignal transduction pathway activity analysis in pancreatic ductal adenocarcinoma in search for new therapeutic targets
Addition of graphical abstract
Abstract:
Line 27: publicly available
Lines 36-37: This is the first time that STP activity has been quantified in PDAC.
Keywords:
We have removed ’pancreatic cancer’ as keyword, as pancreatic ductal adenocarcinoma has already been mentioned. STAP was written out as simultaneous transcriptome-based activation profiling and added as keyword. In addition, ‘MAPK’ was added as important STP in PDAC.
Keywords were reordered to mention: STPs, disease, technique, therapy.
Introduction:
Line 69: publicly available
Lines 72-73: Alterations in STP activity may drive oncogenesis play a role in tumor growth and metastasis and pose opportunities for therapeutic interventions.
Results:
Edited Figures 1-6
Line 117 added: (Figure 2A)
Line 119 changed to: Figure 2BA
Line 122 changed to: (Figure 2CB).
Lines 149-152: Figure 2. (A) Schematic interpretation of the PI3K pathway. When PI3K is activated, FOXO activity is blocked and FOXO translocates to the cytoplasm. Oxidative stress induces alternative activation of FOXO with the function to protect against reactive oxygen species (ROS). Schematic representation on how PI3K STP activity is determined from FOXO activity and SOD2 expression.
Line 201 added: altered
Discussion:
Line 414: (ref 41).
Lines 414-416: High activity of the developmental pathway TGFβ may play a causal role in causing epithelial-mesenchymal transition (EMT) due to the high prevalence of SMAD gene mutations in PDAC [18,37].
Lines 423 added: (Figure 2A)
Lines 431-434: Our study relies on these previously generated datasets, which likely differ in methodology and quality and STP activity should be confirmed in future experiments when testing drugs interfering on these pathways.
Lines 440-442: A limitation of the deconvolution software is that it is based RNA expression, which may differ from protein expression.
Lines 460-462: STP activity of cell lines should be verified with consistent culturing conditions and the influence of serum concentration in culture medium should be evaluated.
Lines 463-472 in the discussion: STAP-STP analysis provides unique insights into the abnormal functioning of signal transduction pathways in PDAC. pathways that drive oncogenesis and disease progression of PDAC. MAPK and PI3K are probably STPs that drive tumor growth given that more than 90% of PDAC tumors have KRAS mutations. TGFβ may be another important pathway, given the high mutation rate in SMAD genes [37]. Interfering on specific STPs and quantifying the influence on all STPs provides evidence which STPs drive tumor growth and how these pathways crosstalk. Correlating STP activity to clinical outcomes, such as disease-free interval and overall survival, increases the probability that these pathways are involved in proliferation and strengthens the rationale to interfere in these pathways.
Lines 474-476: This is the first time STP activity has been quantified for PDAC, as this Activity of STPs cannot be performed quantified with conventional bioinformatic analysis methods, as has been described before [9]
Line 481 added: aberrant
Lines 486-489: STAP-STP technology is unique in monitoring the influence of drugs on all STPs and identify crosstalk resulting in therapy resistance, and helping to develop combinational therapy. It also STAP-STP technology uniquely enables development and evaluation of cancer differentiation therapy by affecting all relevant pathways.
Methods:
Updated the references in Table 2
Lines 530-531: Activity of Forkhead box O (FOXO) transcription factor
Lines 534-535 :superoxide dismutase 2 (SOD2)
Line 536 added: (Figure 2A)
References:
Checked and updated references
Reference 37 was added: Dardare, J.; Witz, A.; Merlin, J.L.; Gilson, P.; Harle, A. SMAD4 and the TGFbeta Pathway in Patients with Pancreatic Ductal Adenocarcinoma. Int J Mol Sci 2020, 21, doi:10.3390/ijms21103534.

Round 2
Reviewer 2 Report
Comments and Suggestions for Authors
Accept in present form